# Histone H3 threonine 11 phosphorylation by Sch9 and CK2 regulates chronological lifespan by controlling the nutritional stress response

Seunghee Oh, Tamaki Suganuma*, Madelaine M Gogol, Jerry L Workman*

Stowers Institute for Medical Research, Kansas City, United States

**Abstract** Upon nutritional stress, the metabolic status of cells is changed by nutrient signaling pathways to ensure survival. Altered metabolism by nutrient signaling pathways has been suggested to influence cellular lifespan. However, it remains unclear how chromatin regulation is involved in this process. Here, we found that histone H3 threonine 11 phosphorylation (H3pT11) functions as a marker for nutritional stress and aging. Sch9 and CK2 kinases cooperatively regulate H3pT11 under stress conditions. Importantly, H3pT11 defective mutants prolonged chronological lifespan (CLS) by altering nutritional stress responses. Thus, the phosphorylation of H3T11 by Sch9 and CK2 links a nutritional stress response to chromatin in the regulation of CLS.
DOI: https://doi.org/10.7554/eLife.36157.001

## Introduction

Nutritional stress is an unavoidable event for most organisms, and appropriate metabolic adaptation to nutritional deficiency is essential to ensure the survival of cells and organisms. Since calorie restriction may modulate genetic or metabolic switches implicated in longevity from yeast to mammals (*Guarente, 2006*), proper metabolic adaptation due to nutritional changes may be critical processes in the regulation of lifespan. *Saccharomyces cerevisiae* utilizes different carbon sources and adapts to various nutritional environments by changing its metabolism (*Broach, 2012*). In yeast, glucose is the preferred carbon source for growth. When external glucose levels are sufficient, yeast cells utilize fermentation for energy production even if the oxygen concentration is high. When the levels of glucose and other fermentable carbon sources run low, they shift energy metabolism from fermentation to the mitochondrial respiration pathway. Multiple signaling pathways including PKA/RAS, TOR, Sch9 cooperate to regulate this metabolic transition (*Galdieri et al., 2010*; *Wilson and Roach, 2002*), which is accompanied by global changes in gene expression (*DeRisi et al., 1997*). Many factors important for regulation of the metabolic transition are also involved in the process of cellular aging (*Cheng et al., 2007*). Downregulation of the TOR, Sch9, and PKA/RAS pathways leads to extension of CLS (*Fabrizio et al., 2001*; *Longo, 1999*; *Powers et al., 2006*; *Wei et al., 2008*), which measures the length of time that non-dividing yeast cells survive (*Longo et al., 2012*).

Chromatin-modifying enzymes also play roles in aging (*Benayoun et al., 2015*; *Sen et al., 2016*). The sirtuin deacetylase Sir2 regulates replicative lifespan (RLS) by reducing histone acetylation levels at telomeres and rDNA regions (*Dang et al., 2009*; *Imai et al., 2000*; *Kaeberlein et al., 1999*). Inactivation of a chromatin remodeling protein, Isw2, extends RLS by induction of genotoxic stress response genes (*Dang et al., 2014*). However, direct connections between nutrition sensing pathways and chromatin regulation in the aging process are still unknown. Interestingly, pyruvate kinases in yeast and humans have been shown to phosphorylate H3 at T11 (*Li et al., 2015*; *Yang et al., 2012*), suggesting that H3pT11 mediates a connection between metabolism and chromatin. Several

*For correspondence:
tas@stowers.org (TS);
jlw@Stowers.org (JLW)

different kinases are responsible for H3pT11. In yeast, Mek1 directly regulates H3pT11 during meiosis (*Govin et al., 2010*; *Kniewel et al., 2017*). In humans, protein kinase N1, PKN1, phosphorylates H3T11 at promoters of androgen receptor dependent genes (*Metzger et al., 2008*), and checkpoint kinase 1, Chk1, phosphorylates H3T11 in mouse embryonic fibroblast cells (*Shimada et al., 2008*).

The casein kinase two complex, CK2, is a ubiquitous serine/threonine kinase complex which plays roles in cell growth and proliferation. CK2 is a conserved protein complex from yeast to human. Yeast CK2 consists of two catalytic subunits (a1 and a2) and two regulatory subunits (b1 and b2) (*Ahmed et al., 2002*; *Litchfield, 2003*). CK2 phosphorylates many kinds of substrates including histones (*Basnet et al., 2014*; *Cheung et al., 2005*; *Franchin et al., 2017*), and this pleiotropy implies a broad function for CK2 in various biological pathways including glucose metabolism (*Borgo et al., 2017*). Interestingly, deletion of a CK2 catalytic subunit, Cka2, extends CLS in yeast; however, the mechanism of how CK2 regulates CLS is unknown (*Fabrizio et al., 2010*).

Here, we found that upon nutritional stress in yeast, the level of H3pT11 is specifically increased at stress responsive genes and regulates transcription of genes involved in metabolic transition. We also found that Sch9 and Cka1, a catalytic subunit of CK2, are required for the phosphorylation of H3T11 under the stress. Importantly, loss of H3pT11 prolongs CLS by altering the stress response at an early CLS stage, suggesting that H3pT11 by CK2 and Sch9 links nutritional stress to chromatin during in the aging process.

## Results

### H3T11 phosphorylation is increased upon the nutritional stress

Our previous study implied a connection between H3pT11 and glucose metabolism (*Li et al., 2015*), therefore, we examined the relationship between H3pT11 and external glucose levels using an antibody specific to H3pT11 (validated in *Figure 1—figure supplement 1*). Culture media for wild type (WT) cells at early mid-log phase was changed from glucose rich (2%) YPD to YP with different concentrations of glucose (0.02, 0.2, or 2%) for 1 hr. The global levels of H3pT11 showed a clear negative correlation with media glucose levels (*Figure 1A*). When the culture media was shifted from YPD to YP with 3% glycerol (YPglycerol), which is non-fermentable and is a nutritionally unfavorable carbon source for yeast, we also observed robust increases in the levels of H3pT11 (*Figure 1B*). H3pT11 levels were not changed in media containing both glucose and glycerol compared to the 2% glucose condition, and direct addition of glucose was sufficient to suppress the increase of H3pT11 levels found in YPglycerol (*Figure 1—figure supplement 2A*). These data demonstrate that H3pT11 levels are specifically increased in low glucose conditions.

To determine how the genomic distribution of H3pT11 changes upon nutritional stress, we performed ChIP-sequencing (ChIP-seq) of H3pT11 in cells cultured in YPD or YPglycerol conditions. In agreement with the western blots (*Figure 1B*), the total number of H3pT11 peaks increased in YPglycerol conditions compared to YPD (*Figure 1—figure supplement 2B*). We identified 366 genes, whose H3pT11 levels were increased upon this nutritional stress (*Figure 1C*). These genes included hexokinase, *HXK1*, and mitochondrial lactate dehydrogenase, *DLD1*, whose expression are known to increase in low glucose conditions (*Lodi et al., 1999*; *Rodríguez et al., 2001*) (*Figure 1D*). Through GO term analysis of the genes, where H3pT11 levels were changed upon the stress (YPglycerol), we found that the genes with increased H3pT11 levels were highly enriched in aging-related processes, stress responses, and metabolic pathways (*Figure 1E*). H3pT11 levels were decreased at 139 genes (*Figure 1—figure supplement 2C*) in YPglycerol compared to YPD. The genes with reduced H3pT11 levels were involved in fermentation and translation, which are generally repressed in nutritional stress conditions (*Figure 1—figure supplement 2D*). These results show that H3pT11 levels are specifically changed at a group of genes involved in the nutritional stress responses.

### H3T11 phosphorylation regulates transcription upon nutritional stress

The genome-wide distribution of H3pT11 strongly suggests that H3pT11 has a role in regulation of the transcriptional response to nutritional stress. We classified RNA polymerase II (Pol II) regulated genes into three or five groups based on their mRNA expression levels of RNA-sequencing (RNA-seq) in YPglycerol condition and compared H3pT11 levels among those groups (*Figure 2A* and *Figure 2—figure supplement 1A*). H3pT11 levels were mostly enriched in promoter regions. In these

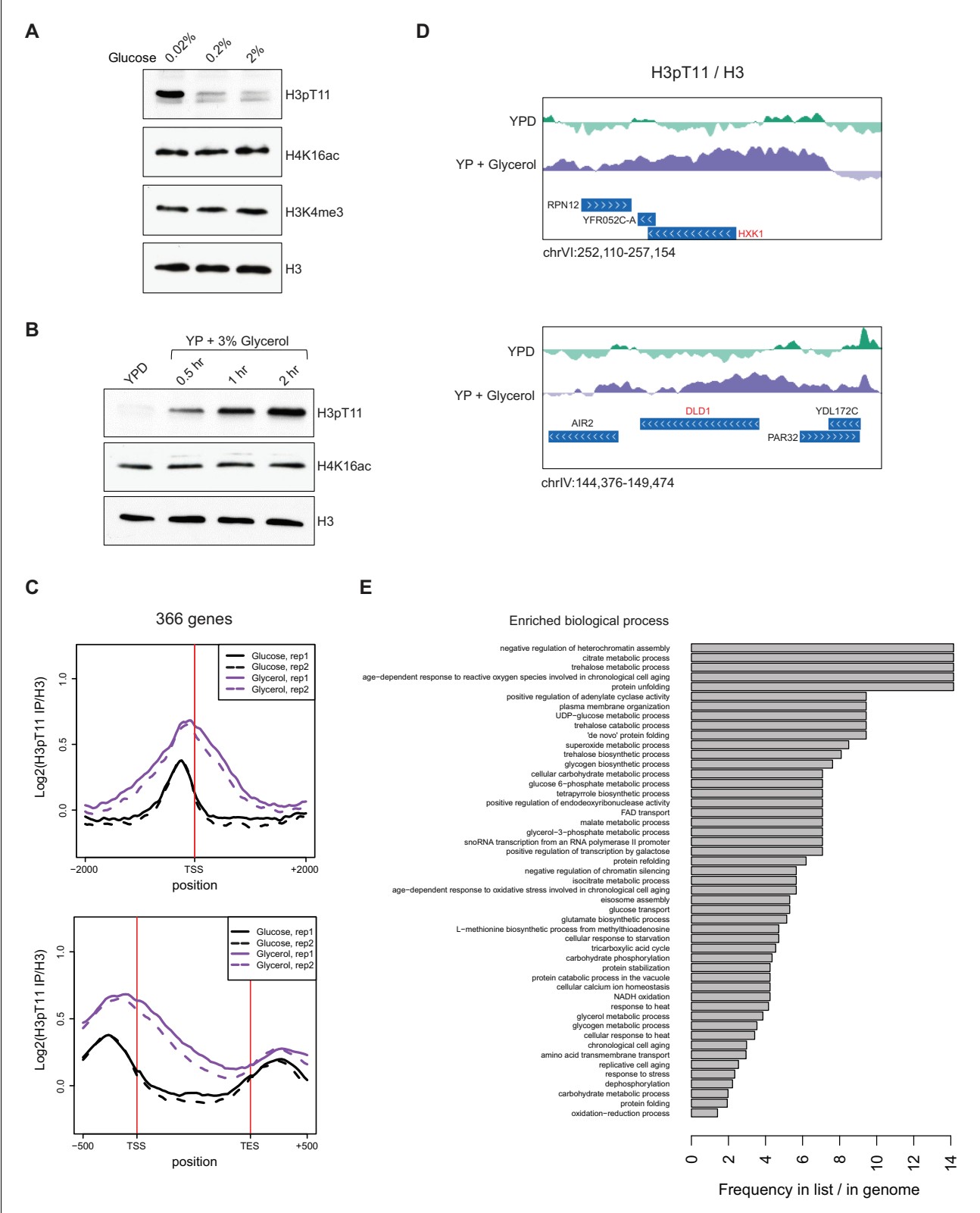

**Figure 1.** H3pT11 responds to nutritional stress. (**A**) H3pT11 levels in the media containing different concentration of glucose measured by western blots. Wild type (BY4741) yeast cultures at early mid-log phase (OD 0.4) were shifted from YPD to YP media containing 0.02, 0.2, or 2% glucose for 1 hr at 30°C. (**B**) H3pT11 levels in the media containing non-fermentable glycerol measured by western blots. WT cultures were shifted from YPD to YP with 3% glycerol for indicated times. (**C**) The averaged profiles of H3pT11 at 366 genes, whose H3pT11 levels are increased in YPglycerol (Glycerol)

*Figure 1 continued on next page*

*Figure 1 continued*

compared to YPD (Glucose) condition. TSS: transcription start site; TES: transcription end site. (D) Normalized H3pT11 levels to H3 at *HXK1* and *DLD1* gene loci in YPD and YPglycerol conditions. (E) GO term analysis of the 366 genes shown in (C). GO terms with a p-value less than 0.05 were included, and with at least two genes from the significant gene list annotated. The bars are based on the frequency of the term in the significant gene list divided by the frequency of the term in the genome.

DOI: https://doi.org/10.7554/eLife.36157.002

The following source data and figure supplements are available for figure 1:

**Source data 1.** H3pT11 occupancy of the genes shown in *Figure 1C* and *Figure 1—figure supplement 2C*.
DOI: https://doi.org/10.7554/eLife.36157.005
**Figure supplement 1.** H3pT11 antibody validation.
DOI: https://doi.org/10.7554/eLife.36157.003
**Figure supplement 2.** H3pT11 responds to low glucose condition.
DOI: https://doi.org/10.7554/eLife.36157.004

regions, the H3pT11 signals were positively correlated with mRNA expression levels in the YPglycerol condition. We compared transcripts in H3T11A mutant to those in WT, cultured in YPD or YPglycerol by RNA-seq. We found a negative correlation of gene expression between YPglycerol dependence (x-axis) and H3T11A dependence (y-axis) with correlation coefficient (cor) −0.38 (*Figure 2B*). Thus, genes with increased expression in YPglycerol tended to be down-regulated in the H3T11A mutant, while genes with decreased expression in YPglycerol were up-regulated in the H3T11A mutant.

As the levels of H3pT11 increased at genes involved in various stress response and aging processes (*Figure 1E*), we analyzed if the transcription of those genes were affected by H3pT11 (*Figure 2—figure supplement 1B*). Chronological aging (cor = −0.8) and oxidative stress response related genes (cor = −0.43) revealed a stronger negative correlation between H3pT11 dependence (y-axis) and YPglycerol dependence (x-axis) compared to the correlation of all genes (cor = −0.38). The genes related to heat response were mostly upregulated upon the stress, and moderately downregulated in H3T11A mutant. When yeast cells encounter a low fermentable carbon source environment, yeast cells switch their metabolism from fermentation to mitochondrial respiration (*Galdieri et al., 2010*). Upon nutritional stress, glucose fermentation pathway genes revealed a strong negative correlation (cor = −0.82) between H3pT11 dependence (y- axis) and YPglycerol dependence (x-axis) (*Figure 2C* upper left panel). Transcripts of the genes related to the TCA cycle and the oxidative phosphorylation pathway were mostly upregulated upon the stress and relatively down-regulated in the H3T11A mutant (*Figure 2C* lower panels). Interestingly, this trend did not match for transcripts of gluconeogenesis specific genes (*Figure 2C* upper right panel). The transcription of these genes was upregulated in the H3T11A mutant, regardless of their transcriptional changes in YPglycerol. As well as carbon source metabolism-involved genes, the transcription of cellular compartment genes was also affected by a H3pT11 defect. Yeast cells contain genes for cytoplasmic and mitochondrial ribosome subunits. Under nutritional stress, the transcription of genes for cytoplasmic ribosome subunits was downregulated, while transcription of genes for mitochondrial ribosome subunits was upregulated (*Figure 2D* upper panel). Interestingly, in the H3T11A mutant, the cytoplasmic ribosome subunit genes were upregulated, while mitochondrial ribosome subunit genes were downregulated in YPglycerol, compared to WT (*Figure 2D* lower panel). Altogether, these data indicated that H3pT11 regulates the transcription of the genes involved in the metabolic transition to the mitochondrial respiratory pathways as well as the genes related to the stress responses upon nutritional stress condition.

## Cka1 is responsible for H3T11 phosphorylation upon nutritional stress

We next asked which kinases are responsible for this modification under nutritional stress conditions. We previously showed that Pyruvate kinase 1 (Pyk1 or Cdc19) in the SESAME (Serine-responsive SAM-containing metabolic enzyme) complex phosphorylates H3pT11 under nutrient rich YPD conditions (*Li et al., 2015*). However, Pyk1 expression is greatly reduced in low glucose conditions (*Boles et al., 1997*). In YPglycerol, we did not observe a clear difference in the global levels of H3pT11 in the SESAME subunit mutants compared to that in WT (*Figure 3—figure supplement 1*). This pointed to a role of different kinase(s) in H3pT11 under YPglycerol. To identify other kinase(s)

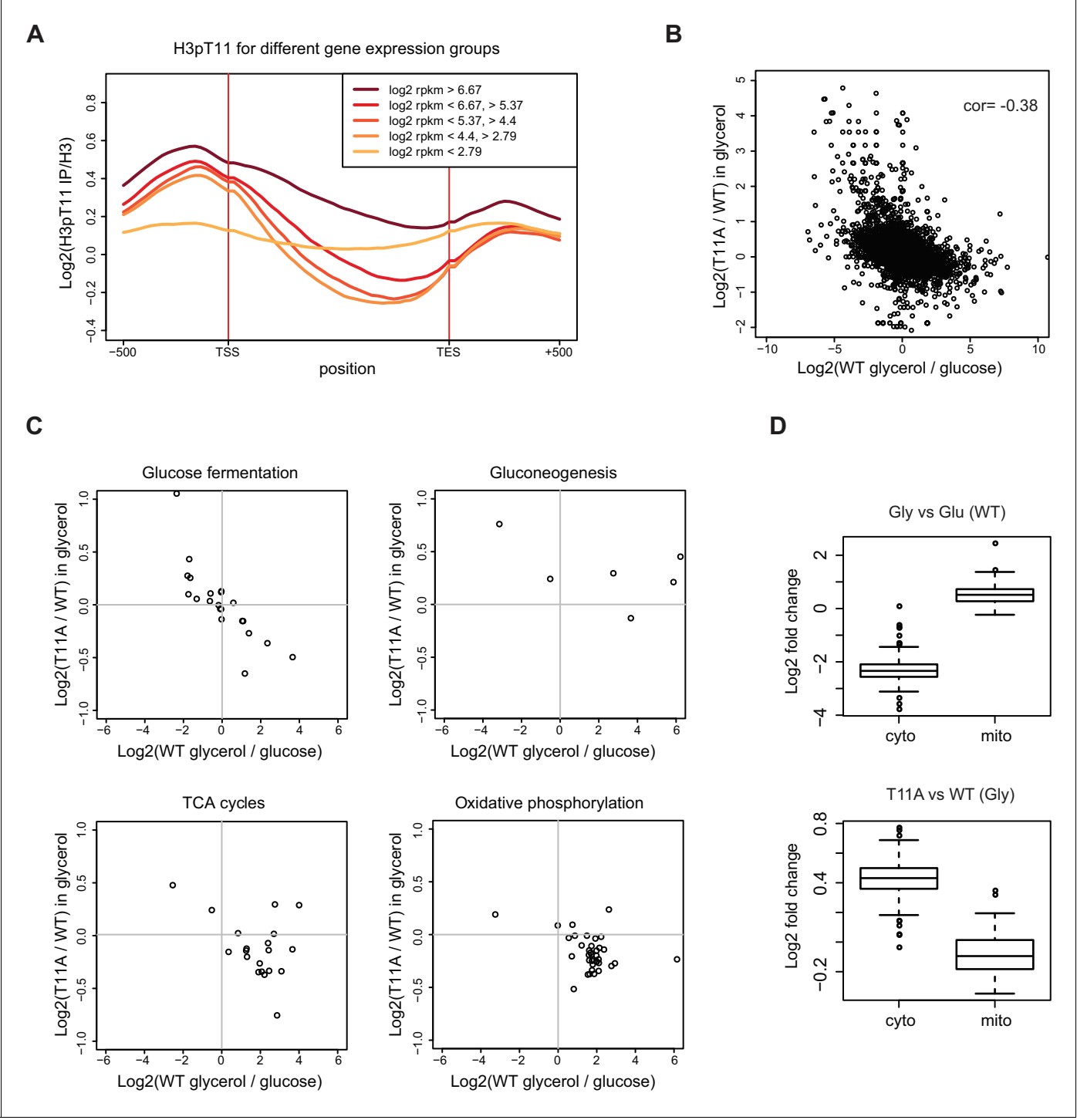

**Figure 2.** H3pT11 regulates transcription involved in metabolic transition upon nutritional stress. (**A**) The average H3pT11 signal of genes from five different gene expression quantiles in YPglycerol. The groups are established by dividing the genes into equally sized quantiles based on the RPKM (Reads Per Kilobase Million) value in YPglycerol. Each group contains 1355 genes. (**B**) A scatter plot from RNA-seq data showing a negative correlation between transcription changes upon media shift from YPD to YPglycerol (x-axis), and the changes in H3T11A mutant compared to WT in YPglycerol condition (y-axis). (**C**) Scatter plots from RNA-seq for transcripts of genes in indicated pathways. (**D**) Box-plots showing expression changes of cytoplasmic (cyto) and mitochondrial (mito) ribosomal subunit genes upon nutritional stress condition (upper panel) and in H3T11A mutant in YPglycerol condition (lower panel). Gly: YPglycerol growth condition; Glu: YPD growth condition.

DOI: https://doi.org/10.7554/eLife.36157.006

The following source data and figure supplement are available for figure 2:

*Figure 2 continued on next page*

*Figure 2 continued*

**Source data 1.** Gene lists analyzed in *Figure 2*.
DOI: https://doi.org/10.7554/eLife.36157.008
**Figure supplement 1.** H3pT11 regulates transcription involved in stress response and aging upon nutritional stress.
DOI: https://doi.org/10.7554/eLife.36157.007

responsible for H3pT11, we tested several kinases including known H3pT11 kinases in yeast and other organisms (*Govin et al., 2010*; *Kniewel et al., 2017*; *Metzger et al., 2008*; *Shimada et al., 2008*). The global levels of H3pT11 were similar among *chk1Δ*, deletion of the yeast homolog of mouse Chk1, *mek1Δ*, and WT (*Figure 3—figure supplement 2A*). Unexpectedly, H3pT11 was decreased in the *cka1Δ* mutant. Interestingly, H3pT11 levels were unaffected upon deletion of another catalytic subunit of CK2, the *cka2Δ* mutant (*Figure 3A* and *Figure 3—figure supplement 2A*), although Cka1 and Cka2 have been thought to be functionally redundant (*Chen-Wu et al.,*

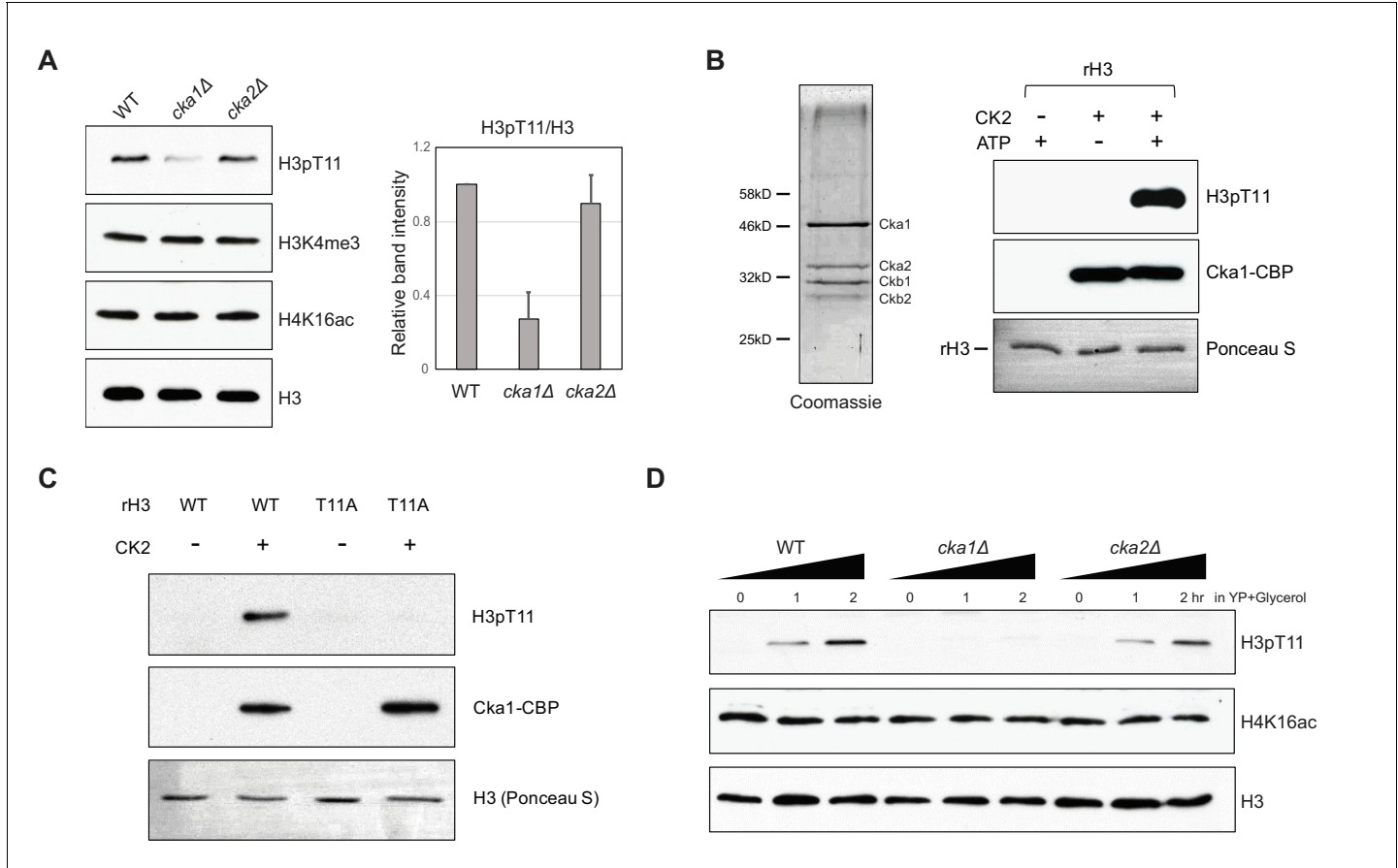

**Figure 3.** Cka1 in the CK2 complex phosphorylates H3T11. (**A**) (Left) H3pT11 levels of WT, *cka1Δ*, and *cka2Δ* in YPD analyzed by western blots. (Right) The relative band intensities of H3pT11 to H3 signals compared to WT. Error bars represent standard deviation (STD) from three biological replicates (**B**) (Left) Coomassie staining of TAP purified CK2 complex using a Cka1-TAP strain. (Right) In vitro kinase assay using TAP purified CK2 and recombinant H3 as a substrate. (**C**) In vitro kinase assay of TAP purified CK2 using recombinant H3 WT or H3T11A mutant as a substrate. (**D**) H3pT11 level changes in WT, *cka1Δ*, and *cka2Δ* mutants upon culture shift to YPglycerol media.
DOI: https://doi.org/10.7554/eLife.36157.009

The following figure supplements are available for figure 3:

**Figure supplement 1.** SESAME is not responsible for increased H3pT11 during nutritional stress condition.
DOI: https://doi.org/10.7554/eLife.36157.010
**Figure supplement 2.** Cka1 is required for H3pT11.
DOI: https://doi.org/10.7554/eLife.36157.011

*1988*; *Padmanabha et al., 1990*). Deletion of the regulatory subunits, *ckb1Δ* and *ckb2Δ*, also did not affect H3pT11 levels (*Figure 3—figure supplement 2B*). Thus, we examined whether the CK2 complex phosphorylated H3T11 by an in vitro kinase assay using a TAP-purified CK2 complex, recombinant H3 (rH3), and ATP. The purified CK2 complex clearly phosphorylated H3T11 (*Figure 3B*). We also confirmed the substrate specificity of CK2 phosphorylation at H3T11 as seen in no-signals on purified recombinant H3T11A mutant by an in vitro kinase assay (*Figure 3C*). Importantly, when we measured the global levels of H3pT11 in YPglycerol, the H3pT11 levels were significantly reduced in *cka1Δ* mutant compared to WT or *cka2Δ* mutant (*Figure 3D*), indicating that Cka1 is responsible for the phosphorylation of H3T11 upon nutritional stress.

## Sch9 regulates H3T11 phosphorylation upon nutritional stress

Since H3pT11 levels respond to glucose levels in the media (*Figure 1*), we further examined whether H3pT11 level is related to glucose-sensing pathways. The Sch9, PKA, and TOR pathways are responsible for glucose sensing in the context of calorie restriction (*Powers et al., 2006*). Sch9, Ras2, and Tor1 proteins are key enzymes in each pathway. To ask whether these pathways were involved in the H3pT11 regulation, we compared the H3pT11 levels among *sch9Δ*, *ras2Δ*, *tor1Δ* mutants, and WT in YPglycerol. Interestingly, H3pT11 increases were significantly impaired only in the *sch9Δ* mutant but not in the *ras2Δ* or *tor1Δ* mutants cultured in YPglycerol (*Figure 4A*), suggesting H3pT11 levels depended on the Sch9 pathway. In support of this observation, we found that TAP-purified Sch9 protein could directly phosphorylate H3T11 in vitro (*Figure 4B*). As both CK2 and Sch9 were responsible for H3pT11 in vivo (*Figures 3A* and *4A*) and were able to phosphorylate H3T11 in vitro (*Figures 3B* and *4B*), we examined if CK2 and Sch9 phosphorylate H3T11 in a cooperative or independent manner in vivo. We measured the global levels of H3pT11 in *sch9Δcka1Δ* double mutant in YPglycerol. Interestingly, the *sch9Δcka1Δ* double mutant showed similar H3pT11 level compared to *cka1Δ* alone (*Figure 4C*). Thus, Sch9 and CK2 are not independent of each other but have overlapping function in regulation of H3pT11 upon nutritional stress. In addition, we found that both Cka1 and Sch9 are bound to chromatin during glucose starvation (*Figure 4D*), suggesting Sch9 and CK2 phosphorylate H3T11 at chromatin under nutritional stress.

## H3T11 phosphorylation regulates CLS

Glucose-sensing pathways are closely related to lifespan control from yeast to mammals (*Cheng et al., 2007*), and deletion of Sch9 results in a well-known long-lived mutant in yeast (*Fabrizio et al., 2001*). H3pT11 was tightly controlled by media glucose levels (*Figure 1*), and Sch9 is responsible for H3T11 phosphorylation (*Figure 4*). We therefore asked whether H3pT11 is involved in the regulation of CLS. Strikingly, CLS was significantly extended in the H3T11A mutant compared to the WT strain (*Figure 5A*). In addition, a *cka1Δ* mutant also extended CLS (*Figure 5B*). However, the *sch9Δcka1Δ* mutant did not further extend the CLS of the *cka1Δ* or *sch9Δ* single mutants (*Figure 5C* and *Figure 5—figure supplement 1*). These data suggest that Sch9 and CK2 cooperatively regulate CLS as seen in the H3pT11 regulation.

Sch9 and CK2 might regulate the CLS by controlling the phosphorylation of H3T11 in response to alteration of glucose levels during chronological aging. To address this hypothesis, we tracked H3pT11 levels during the CLS assay. Interestingly, global levels of H3pT11 were significantly increased at day one after inoculation and then reduced (*Figure 5D*). At this time point, media glucose has been consumed and depleted (*Figure 5E*), and yeast cells begin to change the utilization of its carbon source metabolism from fermentation to respiration (*DeRisi et al., 1997*; *Galdieri et al., 2010*). We observed that this process is regulated by H3pT11 in nutritional stress conditions (*Figure 2*). Supplying glucose at day one after inoculation suppressed the elevation of H3pT11 levels (*Figure 5F*). These data suggest that the increase in H3pT11 levels at early stage of CLS is anti-correlated with glucose availability in the media, and H3T11 phosphorylation mediated by Sch9 and CK2 affects lifespan by regulating the metabolic transition at this time point.

## H3pT11 controls CLS by regulation of acetic acid stress response

Upon depletion of glucose, yeast cells encounter several stresses including media acidification. Media acidification, especially by acetic acid produced during the early stage of the CLS assay has been suggested as a pro-aging factor. The glucose sensing pathway via Sch9 is responsible for the

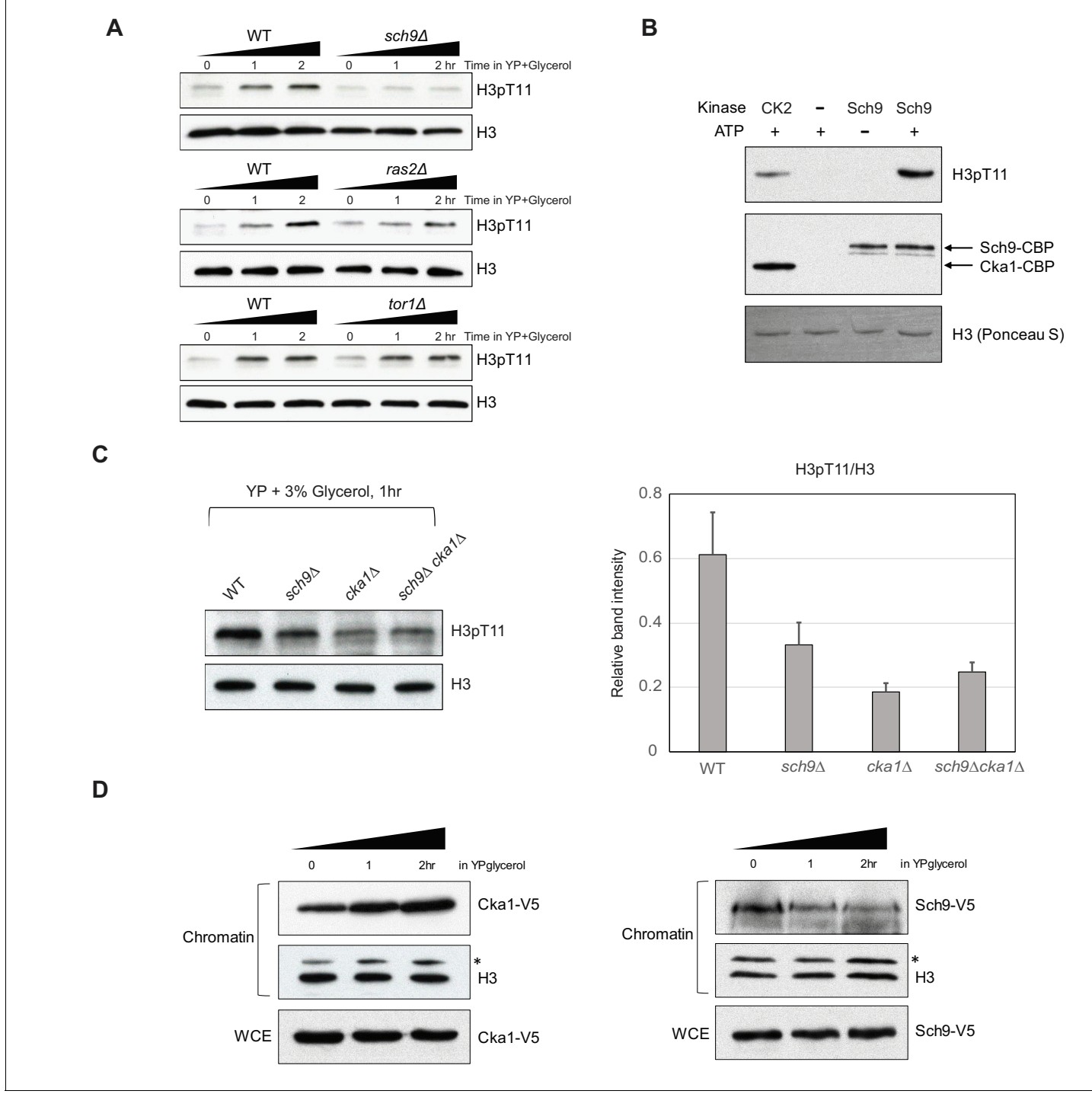

**Figure 4.** Sch9 regulates H3pT11 upon nutritional stress. (**A**) H3pT11 levels in WT, *sch9Δ*, *ras2Δ*, and *tor1Δ* mutants upon media shift to YPglycerol measured by western blots. (**B**) In vitro kinase assay of TAP purified (Sch9-TAP) Sch9 and CK2 using recombinant H3 as a substrate. (**C**) (Left) H3pT11 levels in WT, *sch9Δ*, *cka1Δ*, and *sch9Δcka1Δ* at 1 hr after media shift from YPD to YPglycerol analyzed by western blots (Right) The relative ratios of H3pT11 to H3 signals are presented with error bars indicating STD from three biological replicates. (**D**) The levels of chromatin bound Cka1 (Left) or Sch9 (Right) upon media shifting to YPglycerol measured by western blots. Asterisks (*) indicate nonspecific bands.

DOI: https://doi.org/10.7554/eLife.36157.012

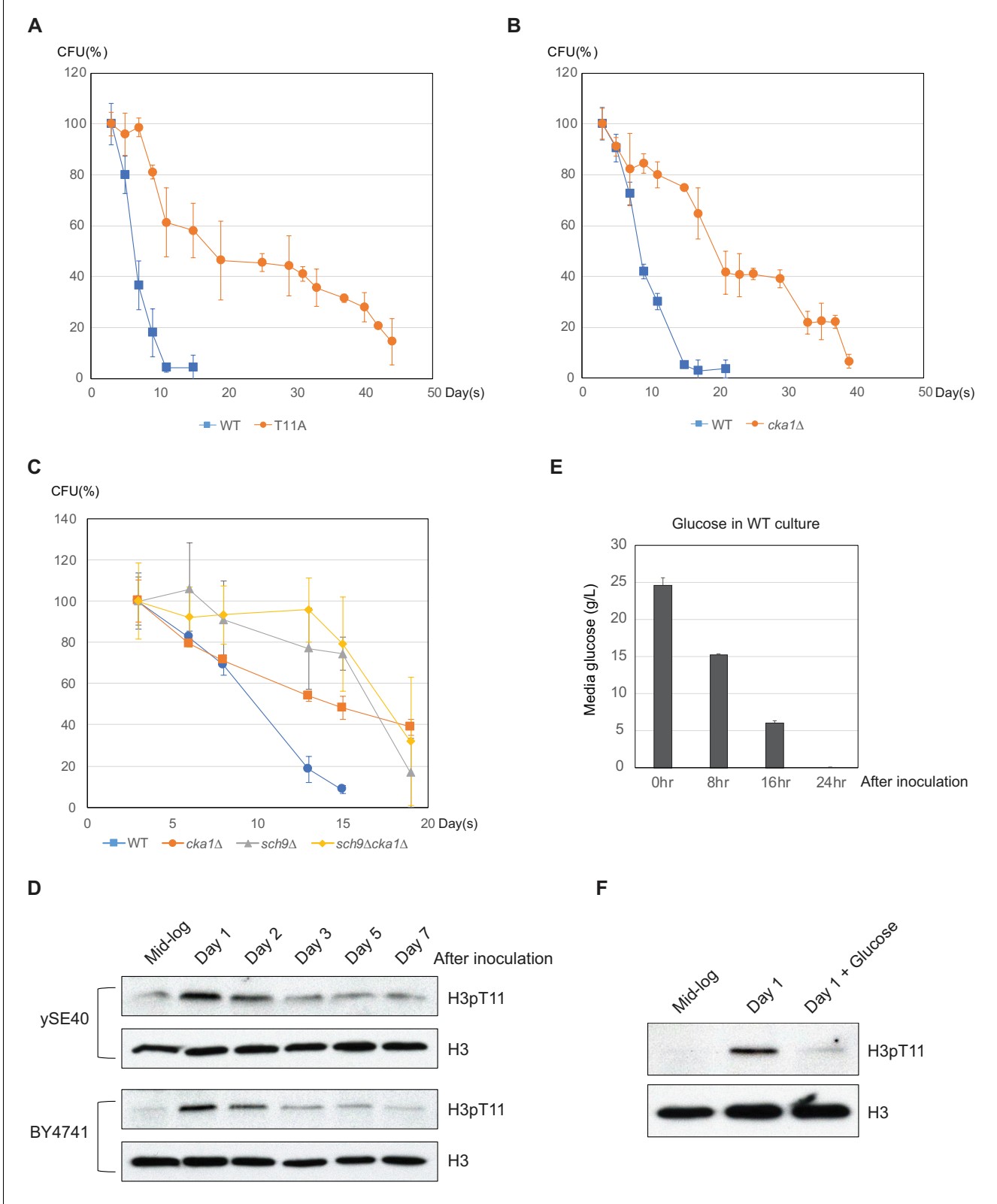

**Figure 5.** Phosphorylation of H3T11 regulates CLS. (A) CLS assays of WT (ySE40) and H3T11A strains. (B) CLS assays of WT (BY4742) and *cka1Δ* strains. (C) CLS assays of WT (BY4742), *cka1Δ*, *sch9Δ*, and *sch9Δcka1Δ* strains. Error bars in CLS assays indicate STD from three biological replicates. CFU: colony- forming units. (D) H3pT11 levels measured at indicated times during CLS assay of ySE40 (WT of histone mutant strains) and BY4741 strain analyzed by western blots. (E) A Bar graph displaying media glucose concentration measured from the WT strain culture at indicated times in the CLS

*Figure 5 continued on next page*

*Figure 5 continued*

assay. Error bars indicate STD of three biological replicates. (**F**) H3pT11 levels of WT strain at exponential growth stage (mid-log), saturated day one culture (day 1), and day one culture with re-supplemented glucose (day 1 + glucose) analyzed by western blots. For day 1 + glucose culture, 2% glucose was directly added to saturated day one culture, then incubated for additional 1 hr.

DOI: https://doi.org/10.7554/eLife.36157.013

The following source data and figure supplement are available for figure 5:

**Source data 1.** CFU values in the CLS assays shown in *Figure 5A, B and C*.

DOI: https://doi.org/10.7554/eLife.36157.015

**Figure supplement 1.** Relative viability of WT (BY4742), *sch9Δ*, *cka1Δ*, and *sch9Δcka1Δ* strains during CLS assay at indicated time points.

DOI: https://doi.org/10.7554/eLife.36157.014

acetic acid stress response. (*Burtner et al., 2009*; *Fabrizio et al., 2005*; *Longo et al., 2012*). To know whether impaired H3pT11 affects acidification or the levels of acetic acid in the media, we measured media acetate level and pH during CLS analysis. There were no significant differences in media pH levels between WT and H3T11A during the first few days of CLS, and the media acetate levels were slightly higher in H3T11A mutant (*Figure 6—figure supplement 1A and B*). However, H3T11A, and *cka1Δ* mutants, as well as *sch9Δ* mutant displayed strong resistance against a high acetic acid concentration in the media (*Figure 6A and B*). In addition, the *sch9Δcka1Δ* mutant showed similar acetic acid resistance to *sch9Δ* single mutant (*Figure 6B*), as in the case of CLS control (*Figure 5C*). We thought that extended CLS in H3T11 phosphorylation defective mutants (H3T11A, *cka1Δ*, *sch9Δ*, and *sch9Δcka1Δ*) might be correlated to their resistance to acetic acid. Supporting this idea, buffering media to pH 6.0 abolished the extension of CLS in H3T11A or *cka1Δ* (*Figure 6C* and *Figure 6—figure supplement 1C*). High level of acetic acid in the media disrupts glucose metabolism (*Sousa et al., 2012*). We observed that media glucose levels remained stable (i.e. glucose was not consumed) even after 24 hr of WT culture in 50 mM acetic acid containing SDC media (*Figure 6—figure supplement 1D*), suggesting impairment of glucose utilization after acetic acid treatment. By contrast, media glucose was consumed in the 10 mM acetic acid condition, similar to the physiological acetic acid concentration of the media during CLS analysis (*Figure 6—figure supplement 1D*) (*Longo et al., 2012*). As H3pT11 responds to low glucose (*Figure 1*), we tested whether media acetic acid affects H3pT11 level. Indeed, H3pT11 was induced by 50 mM acetic acid treatment, but not by 10 mM acetic acid treatment (*Figure 6—figure supplement 1E*). In 50 mM acetic acid, the H3pT11 increase was impaired in *cka1Δ*, *sch9Δ*, and *cka1Δsch9Δ* mutants (*Figure 6D and E*). These data indicate that Sch9 and CK2 are required for the common pathway(s) regulating acetic acid and low glucose stress responses. Altogether, H3pT11 mediated by CK2 and Sch9 regulate early stage of CLS by controlling nutritional stress responses.

## Discussion

### H3pT11 regulation by Sch9 and CK2 upon nutritional stress

In this study, we showed that H3pT11 specifically increased in nutritional stress conditions. H3pT11 regulates transcription upon nutritional stress and accelerates aging or cell death. The phosphorylation of H3T11 upon nutritional stress depends on the Sch9 and CK2 kinases (Summarized in *Figure 7*). Sch9 has been shown to be activated upon nutritious conditions (*Urban et al., 2007*). Phosphorylation of Sch9 by the Torc1 complex is required for its activity and rapidly disappears within a few minutes upon nutritional stress. We found that Sch9 activity is required for the increase of H3pT11 in nutritional stress conditions wherein Sch9 is thought to be inactive (*Figure 4A*). There is the possibility that phosphorylation of H3T11 by either of CK2 or Sch9 under glucose-rich conditions is required for further induction of H3pT11 during glucose starvation. Although the global level of H3pT11 is very low in nutritious YPD media, both Sch9 (*Figure 4A*) and CK2 (*Figure 3A*) are required for this level of H3pT11 in YPD conditions. We cannot exclude the possibility that different kinase(s) other than Sch9 and CK2 are involved in increased H3pT11 upon the stress. However, H3pT11 induction depends on pre-existing H3pT11 levels or cooperation of CK2 with Sch9. Another possibility is that Sch9 functions in the stress conditions. Previous studies suggest that Sch9 is required for the induction of stress response genes by a Torc1-independent manner, and Sch9

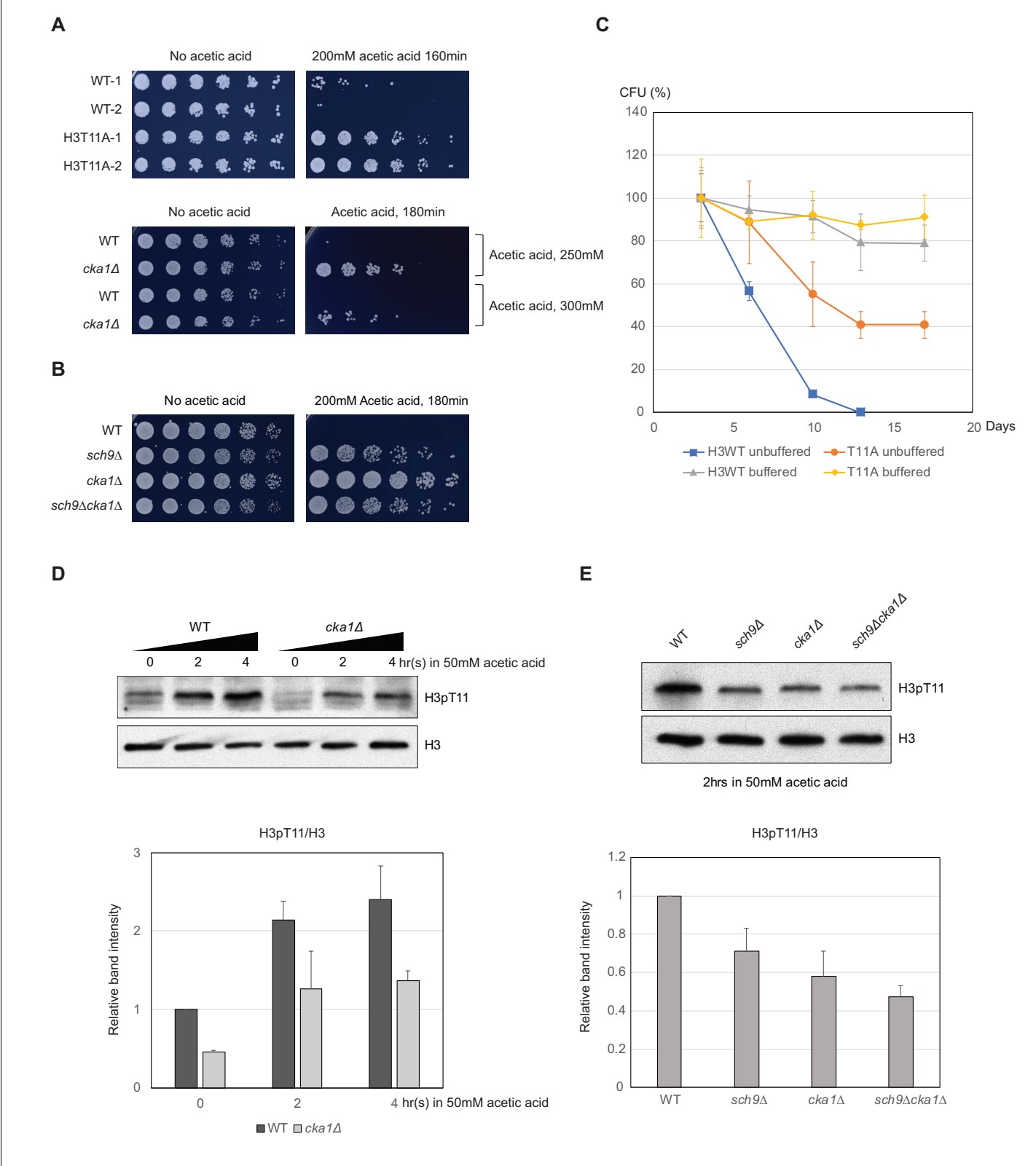

**Figure 6.** H3pT11 affects CLS by regulation of acetic acid resistance. (**A**) Relative viability of H3T11A and *cka1Δ* mutants compared to their WT strains after exposure to indicated durations and concentrations of acetic acid. (**B**) Acetic acid resistance of WT, *sch9Δ*, *cka1Δ*, and *sch9Δcka1Δ*. (**C**) CLS assays of WT and H3T11A strains in buffered (pH 6.0) or unbuffered conditions. (**D**) H3pT11 levels in WT and *cka1Δ* upon 50 mM acetic acid addition analyzed by western blots (upper). The relative band intensities of H3pT11 to H3 signals (lower). (**E**) H3pT11 levels in WT, *sch9Δ*, *cka1Δ*, and *sch9Δcka1Δ* at 2 hr

*Figure 6 continued on next page*

*Figure 6 continued*

after 50 mM acetic acid treatment analyzed by western blots (upper). The relative ratios of H3pT11 to H3 signals (lower). All error bars indicate STD from three biological replicates.

DOI: https://doi.org/10.7554/eLife.36157.016

The following source data and figure supplement are available for figure 6:

**Source data 1.** CFU values in the CLS assays shown in *Figure 6C*.

DOI: https://doi.org/10.7554/eLife.36157.018

**Figure supplement 1.** H3pT11 regulates CLS by modulation of acid stress response.

DOI: https://doi.org/10.7554/eLife.36157.017

activates transcription in response to osmotic stress by its binding to chromatin (*Pascual-Ahuir and Proft, 2007*; *Smets et al., 2008*). We also observed that increases in the levels of H3pT11 upon stress is Tor1 independent (*Figure 4A*). Thus, H3pT11 by Sch9 may function in a Tor1-independent pathway, which is governed by the interaction between CK2 and Sch9. In addition, we found that both Cka1 and Sch9 are bound to chromatin during glucose starvation (*Figure 4D*), implying that they have a role at chromatin in this stress condition.

Our data indicate that Sch9 and CK2 genetically interact with each other in the regulation of H3pT11, acetic acid resistance, and CLS (*Figures 4C*, *5C* and *6B*). Interestingly, CK2 has been shown to phosphorylate Sch9 human homologs S6 kinase (*Panasyuk et al., 2006*) and Akt1 (*Di Maira et al., 2005*). Akt1 also phosphorylates CK2 (*Nguyen and Mitchell, 2013*). In addition, Sch9 and CK2 commonly phosphorylate some transcription factors, including the transcription factors Maf1 and Bdp1 in yeast (*Graczyk et al., 2011*; *Lee et al., 2009*, *Lee et al., 2015*). Sch9 has been suggested to bind to Ckb1, one of the regulatory subunits of CK2 (*Fasolo et al., 2011*). These data suggest an intimate relationship between Sch9 and CK2. However, how these two kinases coordinate H3T11 phosphorylation, and how Sch9 and CK2 cooperatively sense nutrient stress needs to be determined.

## Transcription regulation by H3pT11

H3pT11 has been implicated in transcription activation. In a human prostate tumor cell line, H3T11 phosphorylation by PKN1 is required for androgen stimulated gene transcription by facilitating removal of the repressive H3K9 methylation mark (*Metzger et al., 2008*). In mouse MEF cells, H3pT11 by Chk1 kinase is reduced for the DNA damage response. Decreased H3pT11 levels cause repression of gene transcription, and reduction of H3K9 acetylation by GCN5 (*Shimada et al.,*

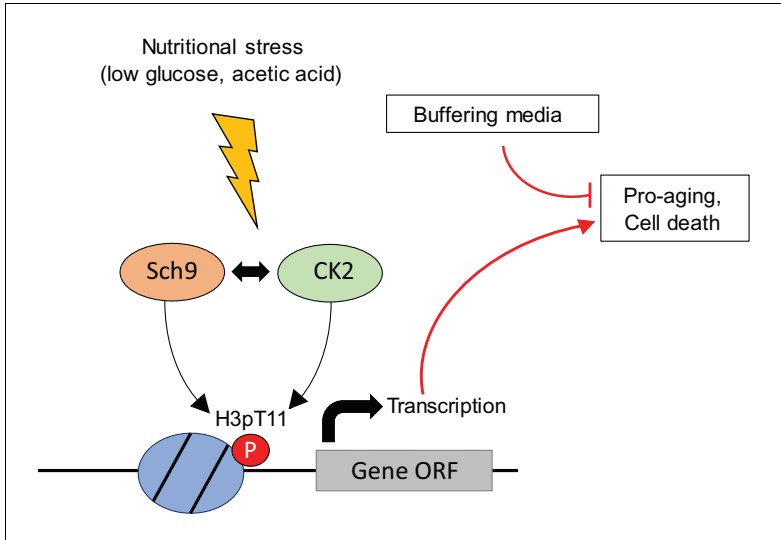

**Figure 7.** Summary models of H3pT11 functions upon stress conditions. Additional files.

DOI: https://doi.org/10.7554/eLife.36157.019

*2008*). We also showed the roles of H3pT11 in transcription regulation (*Figure 2*). However, the mechanism by which H3pT11 regulates transcription in response to nutritional stress is still elusive. Yeast Gcn5 acetyltransferase has been shown to have higher affinity to peptides containing H3S10 phosphorylation (H3pS10) or H3pT11 than unmodified H3 peptides (*Shimada et al., 2008*). Interestingly, the crystal structure of Tetrahymena Gcn5 suggests that H3pS10 may facilitate the interaction between H3pT11 and Gcn5. It has been shown that H3pT11 is required for optimal H3pS10-dependent gene transcription in yeast (*Clements et al., 2003*). Thus, H3pT11 may serve as the binding moiety of chromatin readers such as Gcn5. Crosstalk among H3pT11 and other histone modifications such as H3pS10 may play some role in the regulation of transcription.

## CLS regulation by H3pT11

H3pT11 defective mutants extend CLS (*Figure 5A*) by altering stress response to low levels of glucose (*Figure 2*) and high levels of acetic acid (*Figure 6A*). Adding high levels of acetic acid to the media altered glucose metabolism (*Figure 6—figure supplement 1D*) and increased H3pT11 levels (*Figure 6C*). Although H3pT11 regulatory pathways responding to acetic acid and glucose starvation may be different, both pathways require cooperative Sch9 and CK2 function in the elevation of H3pT11 levels (*Figure 6D*). Intriguingly, while culturing yeast cells in low glucose media prolongs yeast CLS (*Weinberger et al., 2007*), H3pT11, which is induced by low glucose, likely has pro-aging roles (*Figure 5A*). High level of acetic acid can induce apoptosis-like cell death in yeast (*Ludovico et al., 2001*). As seen in strong resistance to acetic acid in H3pT11 defective mutants (*Figure 6A and B*), Sch9 and CK2 dependent H3pT11 induction may facilitate apoptosis under severe stress condition such as aging or high acetic acid. These H3pT11 effects, however, can be overcome by the beneficial effect of calorie restriction or buffering media (*Figure 6C* and *Figure 6—figure supplement 1C*). It is noteworthy that H3pT11 induction upon nutritional stress depends on Sch9 and CK2 (*Figures 3* and *4*), which have pro-aging roles. To understand the role of H3pT11 in regulation of CLS, the contributions of Sch9 and CK2 to the glucose starvation response needs to be elucidated.

## Materials and methods

**Key resources table**

| Reagent type (species) or resource | Designation | Source and reference | Identifiers |
|---|---|---|---|
| Strain, strain background (*S. cerevisiae*) | Strain background: BY4741 | Open Biosystems | Cat#YSC1048 |
| Strain, strain background (*S. cerevisiae*) | Strain background: BY4742 | Open Biosystems | Cat#YSC1049 |
| Strain, strain background (*S. cerevisiae*) | Strain background: YBL574 | *Nakanishi et al. 2008*, PMID: 18622391 | N/A |
| Antibody | Anti H3pT11 | Abcam | Cat# ab5168, RRID:AB_304759 |
| Antibody | Anti H3 | Abcam | Cat#Ab1791 RRID:AB_302613 |
| Antibody | Anti H3K4me3 | EMD Millipore | Cat#07–473 RRID:AB_1977252 |
| Antibody | Anti H4K16ac, Rabbit polyclonal | *Huang et al., 2014* PMID: 25512562 | N/A |
| Antibody | Anti CBP, Rabbit polyclonal | *Venkatesh et al., 2012* PMID: 22914091 | N/A |
| Recombinant DNA reagent | pFA6A-NatMX6 | Euroscarf | Cat#P30437 |
| Recombinant DNA reagent | pFA6a-6xGLY-V5-kanMX6 | Addgene | Cat#20780 |
| Commercial assay or kit | KAPA HTP library prep kit | KAPA Biosystems | Cat#KK8234 |

*Continued on next page*

*Continued*

| Reagent type (species) or resource | Designation | Source and reference | Identifiers |
|---|---|---|---|
| Commercial assay or kit | TruSeq Stranded Total RNA Library Prep Kit with Ribo-Zero Gold set B | Illumina | Cat#RS-122–2302 |
| Commercial assay or kit | HiSeq SR Cluster Kit v4 cBot | Illumina | Cat#GD-401–4001 |
| Commercial assay or kit | HiSeq SBS Kit v4 (50 cycles) | Illumina | Cat#FC-401–4001 |
| Commercial assay or kit | Glucose Colorimetric Detection Kit | ThermoFisher Scientific | Cat#EIAGLUC |
| Commercial assay or kit | Acetate Colorimetric Assay Kit | Sigma-Aldrich | Cat#MAK086 |
| Software, algorithm | bcl2fastq 1.8.4 | Illumina | RRID:SCR_015058 |
| Software, algorithm | bowtie2 (2.2.0) | PMID: 22388286 | RRID:SCR_005476 |
| Software, algorithm | R (3.2.2) | https://www.R-project.org/. | RRID:SCR_001905 |
| Software, algorithm | DiffBind (2.0.9) | PMID: 22217937 | RRID:SCR_012918 |
| Software, algorithm | bedtools (2.26.0) | PMID: 20110278 | RRID:SCR_006646 |
| Software, algorithm | tophat (2.0.13) | PMID: 23618408 | RRID:SCR_013035 |
| Software, algorithm | edgeR (3.14.0) | PMID: 19910308 | RRID:SCR_012802 |
| Software, algorithm | UCSC genome browser | PMID: 26590259 | RRID:SCR_005780 |

## Yeast strains

All yeast strains used in this study are described in *Supplementary file 1*. All single deletion mutants using the KanMX4 marker and TAP tagged strains using the HIS3 marker derived from BY4741 and BY4742 were obtained from Open Biosystems library (maintained at the Stowers Institute Molecular Biology facility). Histone H3 mutant shuffle strains were generated and maintained by Stowers Institute Molecular Biology facility (*Nakanishi et al., 2008*). Further deletion from these strains was achieved by targeted homologous recombination of PCR fragments containing marker genes flanked by the ends of the targeted genes. These strains were confirmed by PCR with primer set specific for their deletion marker or coding regions.

## Yeast culture conditions

Overnight saturated cell cultures were diluted into fresh YPD media and incubated until early mid-log phase. For nutritional stress experiments, these cultures were pelleted and washed once with YP containing no carbon source. Washed pellets were resuspended with YP media containing various concentrations of glucose as described in *Figure 1A* or 3% glycerol elsewhere, and incubated at 30°C. For 'glucose added' samples in *Figure 1—figure supplement 2A* and *Figure 5F*, 2% glucose was directly added to the culture in YPglycerol (*Figure 1—figure supplement 2A*) or day one culture in SDC (*Figure 5F*) for 1 hr, at 30°C. For the *cdc19-1* temperature sensitive mutant in *Figure 3— figure supplement 1A*, the cells were cultured in YPD at 25°C and were then shifted to YP-glycerol media at 37°C.

## Preparation of yeast whole cell extracts

Yeast whole cell extracts were prepared as previously described with minor modifications (*Li et al., 2015*). 5 OD (optical density) units of cells were taken from 10 to 15 mL cultures. Harvested cell pellets were transferred to 1.7 mL Eppendorf tubes and washed once with 1 mL distilled water. Cell pellets were resuspended in 250 μL of 2M NaOH with 8% β-Mercaptoethanol and incubated on ice for 5 min. Cells were pelleted and washed once with 250 μL TAP extraction buffer (40 mM HEPES pH 7.5, 10% Glycerol, 350 mM NaCl, 0.1% Tween-20, phosphatase inhibitor cocktail from Roche, and proteinase inhibitor cocktail from Roche). Pelleted cells were resuspended in 180 μL 2X SDS sample buffer and boiled at 100°C for 5 min. 10 μL of each sample was used per lane for western blotting.

## Chronological life span (CLS) assay

CLS assays were performed as described previously with minor modifications (*Longo et al., 2012*). Saturated cultures in SDC media were diluted into fresh unbuffered SDC media or SDC media buffered at pH 6.0 by citrate-phosphate buffer (*Burtner et al., 2009*). Cultures were incubated at 30°C with 220 rpm shaking for aeration. At indicated times, the same number of cells, based on optical density, were taken and plated into fresh YPD plate. The grown colony numbers were counted 2 days after the plating. Colony Forming Units (CFUs) were calculated by dividing the number of colonies grown at each time point by the number of colonies at day 3 (set as 100%).

## Yeast chromatin fractionation

Yeast chromatin was fractionated as previously described (*Keogh et al., 2005*) with modifications. 40 to 50 OD cells were collected and washed with SB buffer (1M Sorbitol, 20 mM Tris pH 7.5). Pellets were successively washed with PSB (20 mM Tris pH 7.5, 100 mM NaCl, 20 mM EDTA, 10 mM β-Mercaptoethanol), SB buffer then digested with 1 mg/mL Zymolyase (SEIKAGAKU) in SB buffer for 1 hr at RT. Spheroplasts were washed with SB, then lysed in ice by 0.5% EBX (20 mM Tris pH 7.5, 100 mM NaCl, 15% β-Mercaptoethanol, 0.5% Triton X-100). Lysates with 0.5% EBX were layered by NIB buffer (20 mM Tris pH 7.5, 100 mM NaCl, 15% β-Mercaptoethanol, 1.2M Sucrose), then fractionated by centrifugation at 12,000 rpm, 4°C for 15 min. Pellets were resuspended and lysed again in 1% EBX (20 mM Tris pH 7.5, 100 mM NaCl, 15% β-Mercaptoethanol, 1% Triton X-100). Lysates were fractionated by centrifugation at 14,000 rpm, 4°C for 10 min. Pellets were resuspended with 2X sample buffer and boiled at 100°C for 5 min.

## Chromatin IP

Chromatin IP assays were performed as previously described with minor modifications (*Shim et al., 2012*). 100 mL cultures were subjected to crosslinking by addition of 3 mL of 37% formaldehyde (Sigma) at RT for 15 min with constant swirling. 6 mL of 2.5M glycine was added to quench crosslinking reaction at RT for 5 min. 80 OD quenched cells were pelleted by centrifugation at 6000 rpm for 5 min and were washed twice with ice-cold 1X TBS (20 mM Tris pH 7.5 and 150 mM NaCl). Cells were lysed by bead beating in lysis buffer (50 mM HEPES pH 7.5, 150 mM NaCl, 1 mM EDTA, 1% Triton X-100, 0.1% Sodium deoxycholate, and 0.2% SDS). Lysates were sonicated to generate short DNA fragments using a Sonic Dismembrator Model 500 (Fisher) and were then clarified by centrifugation at 12,000 rpm, 4°C for 20 min. Clarified lysates were diluted in four times volume of lysis buffer without SDS containing fresh Complete Mini protease inhibitor cocktail (Roche) and were then subjected to immunoprecipitation with the following antibodies: 4 µL of anti-H3pT11 (ab5168, Abcam) or 2 µL of anti-H3 (ab1719, Abcam). Antibody bound DNA was recovered by incubation with 40 µL protein A agarose beads (GE Healthcare) at 4°C for overnight. Beads were washed sequentially with the following buffers: once with lysis buffer without SDS for 10 min, twice with 500 mM NaCl lysis buffer without SDS for 10 min, once with LiCl buffer (10 mM Tris-HCl pH 8.0, 250 mM LiCl, 1 mM EDTA, 0.5% NP-40, and 0.5% Sodium deoxycholate) for 10 min, and twice with TE buffer (10 mM Tris-Cl pH 7.5 and 1 mM EDTA) for 5 min. The DNA/chromatin complexes were then eluted twice by incubation in elution buffer (1% SDS and 250 mM NaCl) at 65°C for 30 min with occasional vortexing. Eluate was treated with Proteinase K (Sigma) at 55°C for 2 hr and were then incubated at 65°C for overnight to reverse crosslinking. DNA was prepared by phenol/chloroform extraction followed by ethanol precipitation. Precipitated DNA was used for RT-qPCR or making libraries for ChIP-Sequencing.

## RNA purification

Yeast RNAs were prepared as previously described (*Schmitt et al., 1990*). Briefly, 5 OD units of yeast cells were taken from cultures and were washed once with 1 mL of DEPC treated water. Washed pellets were transferred into 1.7 mL Eppendorf tubes and resuspended in 400 µL of AE buffer (50 mM sodium acetate pH5.3 and 10 mM EDTA). 40 µl of 10% SDS was added to AE buffer resuspended cells and vortexed. 440 µL of phenol pH 8.0 (Sigma) was added to tubes, and then tubes were incubated at 65°C for 4 min. Tubes were rapidly cooled down in pre-chilled ice block until phenol crystals appear and were then centrifuged at 11,000 rpm for 2 min. Aqueous phase was

carefully transferred into new tubes. RNAs in the aqueous phase were prepared using phenol/chloro-form extraction followed by ethanol precipitation.

## TAP purification

TAP purification of the CK2 complex was carried out as previously described (*Li et al., 2015*). 6L cultures of the Cka1-TAP strain were grown in YPD medium at 30°C to an OD about 2.0 at 600 nm. The cell pellets were resuspended in TAP extraction buffer (40 mM HEPES pH 7.5, 10% Glycerol, 350 mM NaCl, 0.1% Tween-20, and protease inhibitor cocktail from Roche) and were then disrupted by bead beating. The crude cell extracts were treated with 125 U Benzonase and 50 μL of 10 mg/ml heparin at RT for 15 min to remove nucleic acid contamination and were then clarified by ultracentrifugation. Clarified extracts were incubated with IgG Sepharose (GE healthcare) beads at 4°C for 3 hr. The IgG-beads bound proteins were resuspended in TEV cleavage buffer (10 mM Tris pH 8.0, 150 mM NaCl, 0.1% NP-40, 0.5 mM EDTA, 10% glycerol, and Complete Mini protease inhibitor cocktail from Roche) and cleaved by addition of 5 μl of AcTEV (Invitrogen) at 4°C overnight. The cleaved proteins were resuspended in calmodulin binding buffer (10 mM Tris pH 8.0, 300 mM NaCl, 1 mM magnesium acetate, 1 mM imidazole, 2 mM $CaCl_2$, 0.1% NP-40, and 10% glycerol) and incubated with Calmodulin Sepharose (GE healthcare) beads at 4°C for 4 hr. Calmodulin-resin bound proteins were eluted by resuspension with calmodulin elution buffer (10 mM Tris pH 8.0, 150 mM NaCl, 1 mM magnesium acetate, 1 mM imidazole, 2 mM EGTA, 0.1% NP-40, 10% glycerol, and Complete Mini protease inhibitor cocktail from Roche).

## In vitro kinase assay

10 μL of TAP purified CK2 complex or Sch9 were incubated with 800 ng of recombinant Xenopus histone H3 with or without addition of 10 mM ATP in NEBuffer for protein kinase (NEB) at 30°C for 1 to 3 hr. The reactions were quenched by addition of SDS sample buffer and boiled at 100°C for 5 min.

## Spotting assay for acetic acid resistance

Overnight yeast cultures were diluted and were grown until their optical densities at 600 nm reached mid-log phase. The cultures were treated with 200–300 mM acetic acid for 160–180 min. After the treatment, 4-fold serially diluted cells were spotted onto YPD plates. Plates were incubated at 30°C for 1 to 2 days prior to image capture.

## Media glucose and acetate quantification

Aliquots of yeast cultures were pelleted at indicated times, then supernatants were collected and frozen at −80°C until used. Glucose, and acetate concentrations in the growth medium were measured using enzymatic assay kits (EIAGLUC from ThermoFisher Scientific for glucose, and MAK086 from Sigma for acetate detection.) following manufacturers' protocols.

## ChIP-sequencing and RNA-sequencing

ChIP-seq samples were sequenced in two lanes of an Illumina HiSeq 2500 at 51 bases, single end. Data were converted to fastq and demultiplexed using bcl2fastq. Reads were aligned to UCSC genome sacCer3 using bowtie2 (2.2.0) with option '-k 1'. Downstream analysis was done in R (3.2.2). Peaks were called using a custom perl script, requiring a 2-fold change between ip and input samples extending for 50 bases. Peaks closer than 400 bases were merged. This custom script is publicly available from the following link: https://github.com/mmarchin/cbio.seo.100.git) (*Gogol, 2018*; copy archived at https://github.com/elifesciences-publications/cbio.seo.100). Differential peaks of H3pT11 between glucose and glycerol were called using R package DiffBind (2.0.9). Genes closest to differential peaks were identified using bedtools closest (2.26.0) to identify the closest transcription start site. After removing any Pol III, tRNA, and rRNA genes, 366 peaks were found up in glycerol versus glucose, and 139 peaks were down in glycerol versus glucose. Gene ontology enrichment was performed using a hypergeometric test in R. Terms shown in the barplot had p-value<0.05. The length of the bar represents the fold enrichment of the term's frequency in the list given the frequency of the term in the genome. Metagene plots were generated in R using 101-base mean-smoothed windows ± 2000 bases around the TSS or ± 500 bases around the transcript region (start to end).

Different length genes were accommodated using the approx() function, which uses linear interpolation to define the approximated data points. After getting approximated values for each gene, the mean value at each position was used to generate the plot.

RNA-seq samples were sequenced in two lanes of an Illumina HiSeq 2500 at 51 bases, single end. Data were converted to fastq and demultiplexed using bcl2fastq. Reads were aligned to UCSC genome sacCer3 with annotation from Ensembl 84 using tophat (2.0.13) with '-x 1 g 1'. Reads were counted on genes (unioned exon space) using bedtools coverage (2.26.0). Data were read into R (3.2.2). Differentially expressed genes were found using R package edgeR (3.14.0) and required to have BH-adjusted p-value<0.05 and two-fold change in order to be called differentially expressed. All correlations shown were calculated using cor () function in R, which is Pearson correlation by default. Lists of genes for the cyto/mito boxplot were taken from previous work (*Cheng et al., 2007*).

## Acknowledgements

We thank Dr. Swaminathan Venkatesh and Dr. Michael Church for scientific editing of the manuscript. We thank the Workman Lab members and Stowers core facilities for support during this project. This work was supported by funding from Stowers Institute for Medical Research, and National Institutes of General Medical Sciences grant R35GM118068. Original data underlying this manuscript can be accessed from the Stowers Original Data Repository at http://www.stowers.org/research/publications/LIBPB-1320.

## Additional information

### Competing interests

Jerry L Workman: Reviewing editor, *eLife*. The other authors declare that no competing interests exist.

### Funding

| Funder | Grant reference number | Author |
| --- | --- | --- |
| Stowers Institute for Medical Research | Workman Lab | Jerry L Workman |
| National Institute of General Medical Sciences | NIH R35 GM118068 | Jerry L Workman |

The funders had no role in study design, data collection and interpretation, or the decision to submit the work for publication.

### Author contributions

Seunghee Oh, Conceptualization, Formal analysis, Investigation, Methodology, Writing—original draft; Tamaki Suganuma, Conceptualization, Formal analysis, Supervision, Writing—original draft, Writing—review and editing; Madelaine M Gogol, Software, Formal analysis, Validation, Methodology; Jerry L Workman, Conceptualization, Formal analysis, Supervision, Funding acquisition, Writing—original draft, Writing—review and editing

### Author ORCIDs

Seunghee Oh http://orcid.org/0000-0002-6701-9473
Tamaki Suganuma http://orcid.org/0000-0001-5149-4002
Madelaine M Gogol http://orcid.org/0000-0002-8738-0995
Jerry L Workman http://orcid.org/0000-0001-8163-1952

### Decision letter and Author response

Decision letter https://doi.org/10.7554/eLife.36157.026
Author response https://doi.org/10.7554/eLife.36157.027

## Additional files

### Supplementary files

• Supplementary file 1. Yeast strains used in this study.
DOI: https://doi.org/10.7554/eLife.36157.020

• Transparent reporting form
DOI: https://doi.org/10.7554/eLife.36157.021

### Data availability

Sequencing data have been deposited at NCBI Dataset ID: GSE111219, and original data underlying this manuscript can be accessed from the Stowers Original Data Repository at http://www.stowers.org/research/publications/LIBPB-1320

The following dataset was generated:

| Author(s) | Year | Dataset title | Dataset URL | Database, license, and accessibility information |
|---|---|---|---|---|
| Seunghee Oh, Madelaine Gogol | 2018 | Histone H3 T11 phosphorylation by Sch9 and CK2 regulates lifespan by controlling the nutritional stress response | https://www.ncbi.nlm.nih.gov/geo/query/acc.cgi?acc=GSE111219 | Publicly available at the NCBI Gene Expression Omnibus (accession no: GSE111219) |

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
