## [Decision Letter]

Thank you for submitting your article "Histone H3 T11 phosphorylation by Sch9 and CK2 regulates lifespan by controlling the nutritional stress response" for consideration by *eLife*. Your article has been reviewed by three peer reviewers, including Matt Kaeberlein as the Reviewing Editor and Reviewer #1, and the evaluation has been overseen by Kevin Struhl as the Senior Editor. The following individuals involved in review of your submission have agreed to reveal their identity: Christopher Burtner (Reviewer #3).

The reviewers have discussed the reviews with one another and the Reviewing Editor has drafted this decision to help you prepare a revised submission.

The reviewers all appreciated the novelty and importance of this work. A few significant concerns were raised during the review process that should be addressed in a revised manuscript. Much of this can be addressed through appropriate changes to the text and discussion, although a couple of key experiments were also suggested.

1) The study provides a fairly comprehensive analysis of H3pT11 in chronological aging of yeast cells, but the data for replicatively aged cells is rather limited and diffuse from the rest of the manuscript. Although the authors show changes in H3pT11 during replicative aging using the mother enrichment program, the functional consequences of these changes are unclear. The obvious question is whether H3pT11 influences replicative lifespan and it is surprising the authors would put in the MEP data without that experiment. As is, the current incomplete data will be more confusing than helpful, as some readers may not understand the differences between these two types of aging in yeast. If the authors do not wish to perform the RLS analysis, it would be acceptable to simply remove the data for replicatively aged cells. Related to this, the authors need to be much more careful about clearly indicating whether they are referring to RLS or CLS throughout the entire manuscript. The reviewers recommend never using the word "lifespan" alone and instead explicitly stating whether you are referring to RLS or CLS. This is important because some of the literature referred to in the introduction are relevant only for RLS while other results are relevant only for CLS and most readers will likely not have the background knowledge to differentiate between the two types of aging in yeast.

2) There are some aspects of the work and the model that are a bit counterintuitive and/or which could be developed more thoroughly. Why does H3pT11 occur upon glucose restriction? Presumably this response is selected for some reason. What is the purpose? It is also counterintuitive that Sch9 would become activated toward this substrate, since Sch9 activity is inhibited under glucose restriction. How does this work? The authors don't actually show that Sch9 is phosphorylating H3 in vivo, and although the genetic evidence and in vitro data are consistent with they are also both indirect. Can Sch9 or Cka2 be localized to chromatin during glucose starvation?

3) Related to #2, the gene expression and stress resistance relationship also seems counterintuitive. Under low glucose/nutritional stress, the authors show that H3T11 becomes hyper-phosphorylated and associated gene expression, including increased stress response and metabolic changes. These changes make perfect sense for CR. And they showed that H3pT11 is responsible for these transcriptional changes since H3T11A had the opposite effects. If so, how do the authors explain the enhanced stress resistance and lifespan extension of H3T11A, since it seems to be antagonizing the stress and metabolic response of CR?

4) Are the reported epigenetic responses really a response to "nutritional stress" or a normal part of the diauxic shift, similar to H3K79 methylation (see Young et al., 2017)? It would be useful to see a direct comparison the genes identified by H3pT11 ChIPseq and those involved in the diauxic shift. The authors hint at similarity in Figure 2C, but no statistical analysis is performed between these two subsets. Seems like overlap may be significant, and if so, this is likely an epigenetic mark of the metabolic shift than of any stress response or aging per se.

5) In Figure 1A, it is important to know when cell lysates were collected and how much glucose is present in the cultures at the time of collection. How long were cells cultured in 0.02%, 0.2% and 2% glucose prior to the Western blot? Glucose is quickly depleted by yeast cells as they enter log phase (your 0.02% glucose will be 0% in 2-3 hours meaning the cells will have already entered respiratory metabolism.) In the other conditions, cells may be cycling in the presence of glucose or not, depending on when the lysate was taken. This missing information may be important when considering whether this is an epigenetic mark of the diauxic shift, or something else, like a stress.

6) Figure 2A is particularly confusing and poorly described. What exactly is the 1355 referring to? What is the justification for selecting the 5 groups?

---

## [Author Response]

The reviewers all appreciated the novelty and importance of this work. A few significant concerns were raised during the review process that should be addressed in a revised manuscript. Much of this can be addressed through appropriate changes to the text and discussion, although a couple of key experiments were also suggested. It was also noted that the manuscript suffered from many typographical and grammatical errors and a lack of clarity in places, several of which are detailed below.

We appreciate the reviewers for their comments on the novelty and importance of this work. It is really nice to have informed reviews that help us improve the manuscript, so we also appreciate the reviewers’ expertise. We also thank you for handling our manuscript.

1) The study provides a fairly comprehensive analysis of H3pT11 in chronological aging of yeast cells, but the data for replicatively aged cells is rather limited and diffuse from the rest of the manuscript. Although the authors show changes in H3pT11 during replicative aging using the mother enrichment program, the functional consequences of these changes are unclear. The obvious question is whether H3pT11 influences replicative lifespan and it is surprising the authors would put in the MEP data without that experiment. As is, the current incomplete data will be more confusing than helpful, as some readers may not understand the differences between these two types of aging in yeast. If the authors do not wish to perform the RLS analysis, it would be acceptable to simply remove the data for replicatively aged cells. Related to this, the authors need to be much more careful about clearly indicating whether they are referring to RLS or CLS throughout the entire manuscript. The reviewers recommend never using the word "lifespan" alone and instead explicitly stating whether you are referring to RLS or CLS. This is important because some of the literature referred to in the introduction are relevant only for RLS while other results are relevant only for CLS and most readers will likely not have the background knowledge to differentiate between the two types of aging in yeast.

We removed the RLS data using MEP strains and related text in the revised manuscript as we hope to highlight our finding of the roles of H3pT11 in CLS regulation in this manuscript. We changed the term of ‘lifespan’ to ‘CLS’ or ‘RLS’ throughout the text. We also stated “chronological lifespan” in the title so this will be clear from the onset.

2) There are some aspects of the work and the model that are a bit counterintuitive and/or which could be developed more thoroughly. Why does H3pT11 occur upon glucose restriction? Presumably this response is selected for some reason. What is the purpose?

In this study we asked whether H3pT11 is important for a transcriptional response to nutritional stress.

We show that H3pT11 promotes the transcription of genes involved in the nutritional stress response and aging phenotype (Figure 1 and Figure 2). We think nutritional stress response and metabolic stress in aging employ H3pT11 as a signal. As the reviewer mentioned, there are still some unclear mechanisms upstream of H3pT11. For example, how Sch9 and CK2 coordinate the phosphorylation of H3T11. We discussed these points in the Discussion section of revised manuscript.

It is also counterintuitive that Sch9 would become activated toward this substrate, since Sch9 activity is inhibited under glucose restriction. How does this work? The authors don't actually show that Sch9 is phosphorylating H3 in vivo, and although the genetic evidence and in vitro data are consistent with they are also both indirect. Can Sch9 or Cka2 be localized to chromatin during glucose starvation?

We appreciate the reviewer’s attention on this point. In this study, we showed Sch9 and CK2, which are important in proliferative growth, are required for H3T11 phosphorylation in nutritional stress conditions. In addition, the roles of Sch9 and CK2 in the regulation of H3T11 phosphorylation, chronological life span, and acetic acid resistance are mutually required although each kinase can independently phosphorylate H3T11 in vitro. As we described above, we hope to further address how Sch9 and CK2 cooperate to phosphorylate H3T11 and how nutrition status and these kinases regulate H3pT11. We discussed these points and described our insights in the Discussion section of the revised manuscript. To address the reviewer’s question, we analyzed chromatin bound proteins in cells cultured in YPD and YPglycerol condition. We found that both CK2 and Sch9 are bound to chromatin during glucose starvation (Figure 4D in the revised manuscript). These data suggest that CK2 and Sch9 phosphorylate H3T11 on chromatin during glucose starvation.

3) Related to #2, the gene expression and stress resistance relationship also seems counterintuitive. Under low glucose/nutritional stress, the authors show that H3T11 becomes hyper-phosphorylated and associated gene expression, including increased stress response and metabolic changes. These changes make perfect sense for CR. And they showed that H3pT11 is responsible for these transcriptional changes since H3T11A had the opposite effects. If so, how do the authors explain the enhanced stress resistance and lifespan extension of H3T11A, since it seems to be antagonizing the stress and metabolic response of CR?

We have revised the manuscript to discuss this in the third paragraph of Discussion section. We have shown that the H3T11A mutant and other mutants defective in H3pT11 regulate CLS by increasing acetic acid resistance (Figure 6). In yeast, high levels of acetic acid can induce apoptotic cell death (Ludovico et al., 2001). The levels of H3pT11 increased in the presence of high levels of acetic acid (Figure 6D). Thus, we think that H3pT11 induction may facilitate apoptosis under severe stress conditions such as aging or high acetic acid in a Sch9 and CK2 dependent manner. However, the effects of H3pT11 can be overcome by beneficial influences of calorie restrictions or buffering media (Figure 6C and Figure 6—figure supplement 1C). We will study if Sch9 functions in the stress condition as described in comment #2. If Sch9 has a pro-aging effect in the stress condition as well as in the nutritious condition, a role of H3pT11 in the stress condition is not counterintuitive because H3pT11 may mimic *sch9Δ* in the stress situation.

4) Are the reported epigenetic responses really a response to "nutritional stress" or a normal part of the diauxic shift, similar to H3K79 methylation (see Young et al., 2017)? It would be useful to see a direct comparison the genes identified by H3pT11 ChIPseq and those involved in the diauxic shift. The authors hint at similarity in Figure 2C, but no statistical analysis is performed between these two subsets. Seems like overlap may be significant, and if so, this is likely an epigenetic mark of the metabolic shift than of any stress response or aging per se.

Diauxic shift is characterized by a switching from rapid fermentative growth to slower aerobic respiration upon the depletion of required carbon sources (Galdieri et al., 2010). In this manuscript, we measured nutritional stress response in the low level of glucose. It has been shown that changes in the cells undergoing diauxic shift are similar to the changes in the cells under stress (Gray et al., 2004; Herman, 2002; Werner-Washburne et al., 1993,). Therefore, we think that diauxic shift is a part of the nutritional stress response in this condition. We showed that the levels of H3pT11 is significantly enriched at genes involved in various stress responses and aging processes in Figure 1E. Thus, H3pT11 functions for not only the transcription of the genes regulating metabolic shift, but also transcription of genes regulating stress response and aging. To make this point clear, we edited the text and added new figures showing that the transcriptions of genes regulating responses to heat, oxidative stress, and chronological aging are also modified by defects of H3T11 phosphorylation upon nutritional stress (Figure 2—figure supplement 1B).

H3K79 methylation during diauxic shift, described in Young et al., 2017, BMC genomics 18, 107, has many features, which differ from features of H3pT11. For instance, the global level of H3K79me3 is not changed during the diauxic shift. H3K79 mono- and di-methylation is abolished in quiescent cells; however, H3K79me2 levels are not changed during diauxic shift. In addition, H3K79me3 loci are not significantly changed between log phase cells and quiescent cells, and H3K79me3 is linked to transcription regulation during this stage. However, we observed that H3pT11 levels robustly increased during this stage (Figure 5D). H3pT11 is correlated with transcription regulation involved in metabolic shift and stress responses (Figure 1E and Figure 2). Thus, we think H3pT11 is not only a consequence of diauxic shift like H3K79 methylation.

5) In Figure 1A, it is important to know when cell lysates were collected and how much glucose is present in the cultures at the time of collection. How long were cells cultured in 0.02%, 0.2% and 2% glucose prior to the Western blot? Glucose is quickly depleted by yeast cells as they enter log phase (your 0.02% glucose will be 0% in 2-3 hours meaning the cells will have already entered respiratory metabolism.) In the other conditions, cells may be cycling in the presence of glucose or not, depending on when the lysate was taken. This missing information may be important when considering whether this is an epigenetic mark of the diauxic shift, or something else, like a stress.

We added more detailed procedure in the text and figure legend in the revised manuscript. WT cultures at early mid-log phase (OD 0.4-0.6) were shifted from YPD to YP containing various concentration of glucose for 1 hour. We have also tested the glucose level after 1hr incubation in YP + 0.02% glucose media as below. We found 1hour incubation, in which we analyzed H3pT11 levels shown in Figure 1A, is not enough to consume all glucose in the media containing 0.02% glucose (Author Response Image 1). We can add this data to the manuscript if required.

**Author Response Image 1 respfig1:** <>

6) Figure 2A is particularly confusing and poorly described. What exactly is the 1355 referring to? What is the justification for selecting the 5 groups?

We apologize for making this confusing. 1355 is the number of genes in each group. The groups are based on dividing the genes into equally sized quantiles based on the RPKM value in YPglycerol condition. We clarified this in revised figure legend 2A. As we arbitrarily divide groups into 5, we also divided groups into three (each group contains 2258 or 2259 genes). We found that group numbers do not lead to changing our conclusion that H3pT11 is positively correlated with mRNA expression levels in nutritional stress condition (Figure 2—figure supplement 1A).